# Application of Pinhole Plasma Jet Activated Water against *Escherichia coli*, *Colletotrichum gloeosporioides*, and Decontamination of Pesticide Residues on Chili (*Capsicum annuum* L.)

**DOI:** 10.3390/foods11182859

**Published:** 2022-09-15

**Authors:** Choncharoen Sawangrat, Yuthana Phimolsiripol, Komgrit Leksakul, Sa-nguansak Thanapornpoonpong, Phanumas Sojithamporn, Maria Lavilla, Juan Manuel Castagnini, Francisco J. Barba, Dheerawan Boonyawan

**Affiliations:** 1Department of Industrial Engineering, Faculty of Engineering, Chiang Mai University, Chiang Mai 50200, Thailand; 2Agriculture and Bio Plasma Technology Center (ABPlas), Thai—Korean Research Collaboration Center (TKRCC), Science and Technology Park, Chiang Mai University, Chiang Mai 50200, Thailand; 3Faculty of Agro-Industry, Chiang Mai University, Chiang Mai 50100, Thailand; 4Department of Preventive Medicine and Public Health, Food Science, Toxicology and Forensic Medicine, Faculty of Pharmacy, University of Valencia, 46100 Valencia, Spain; 5Department of Plant and Soil Sciences, Faculty of Agriculture, Chiang Mai University, Chiang Mai 50200, Thailand; 6AZTI, Food Research, Basque Research and Technology Alliance (BRTA), 48160 Derio, Spain; 7Department of Physics and Materials Science, Chiang Mai University, Chiang Mai 50200, Thailand

**Keywords:** pesticides, cold plasma, decontamination, plasma activated water, chili (*Capsicum annuum* L.), pinhole plasma jet

## Abstract

Plasma activated water (PAW) generated from pinhole plasma jet using gas mixtures of argon (Ar) and 2% oxygen (O_2_) was evaluated for pesticide degradation and microorganism decontamination (i.e., *Escherichia coli* and *Colletotrichum gloeosporioides*) in chili (*Capsicum annuum* L.). A flow rate of 10 L/min produced the highest concentration of hydrogen peroxide (H_2_O_2_) at 369 mg/L. Results showed that PAW treatment for 30 min and 60 min effectively degrades carbendazim and chlorpyrifos by about 57% and 54% in solution, respectively. In chili, carbendazim and chlorpyrifos were also decreased, to a major extent, by 80% and 65% after PAW treatment for 30 min and 60 min, respectively. *E. coli* populations were reduced by 1.18 Log CFU/mL and 2.8 Log CFU/g with PAW treatment for 60 min in suspension and chili, respectively. Moreover, 100% of inhibition of fungal spore germination was achieved with PAW treatment. Additionally, PAW treatment demonstrated significantly higher efficiency (*p* < 0.05) in controlling Anthracnose in chili by about 83% compared to other treatments.

## 1. Introduction

Chili (*Capsicum annuum* L.) is a commonly consumed spice worldwide. It is found in different recipes including salads, curries, soups, pastes, dips, and just about everything [1]. The food safety of fruits and vegetables is of great concern around the world. The Thailand Pesticide Alert Network [2] revealed that fruits and vegetables had a higher level of maximum residue limit (MRL) for pesticide residues according to the Thai Agricultural Standard (TAS 9002-2013). According to the fruits and vegetables sampling for pesticide residues in the Thailand market, among the different samples of chili evaluated, all contained pesticide residues. The existing pesticide residues found in chili are carbofuran, cypermethrin, ethion, methomyl, as well as chlorpyrifos and carbendazim [3,4]. Different adverse health conditions, including cancer, effects on reproduction, immune or nervous system, may be induced by pesticide residues [5]. Besides pesticides, a common food-borne pathogen such as *E. coli* which causes several foodborne outbreaks could be also found in chili. This could lead to diarrhea, hemorrhagic colitis, and hemolytic uremic syndrome [6]. Not only does microbial contamination occur, but Anthracnose, caused by the fungus *Colletotrichum gloeosporioides*, has also been considered a major problem in chili [7,8]. This disease leads to a small yield of chili production due to low seed germination [9] and loss as food waste. Furthermore, this disease also resulted in the low quality of chili as reflected by its poor appearance and low capsaicin and oleoresin content [10].

Typical household processing, chemical treatment, and modern technologies have been used for microbial removal and decontamination of pesticides. However, simple household processes (e.g., washing, peeling, blanching, or refrigerating) are insufficient and not highly effective to solve all these issues. Furthermore, sodium hypochlorite (NaClO) is the most commonly used solution for washing fresh produce, but it causes an odor, is potentially carcinogenic [11], and may also cause eye irritation or oropharyngeal, esophageal, and gastric burns [12]. Moreover, household processes are not highly effective for controlling Anthracnose [13]. Thus, alternative technologies, including high-pressure processing (HPP) [14], pulsed electric fields (PEF) [15], ultrasounds (US) [16], UV light [17,18,19,20], and cold plasma [21], have been developed and proposed to substitute the traditional processing. Among them, cold plasma, one of the promising and environmentally friendly technologies, has gained popularity [22].

Cold plasma consists of electrons, positive and negative ions, free radicals, gas atoms, molecules in the ground or excited state, and quanta of electromagnetic radiation [23,24]. The main reactive species produced by general plasma include reactive oxygen species (ROS) and reactive nitrogen species (RNS). In the last few years, the number of research works on cold plasma for food processing has increased which confirms the significance of this field [25]. While plasma technology has been applied for the degradation of pesticide residues [26,27,28,29], non-thermal plasma has also been applied for the decontamination of microorganisms [30,31,32,33]. Plasma activated water (PAW) is formed when plasma is discharged above or below the water surface [34]. The secondary species, including hydrogen peroxide, nitrate ions, nitric acid, nitric oxide radicals, nitrite ions, nitrous acid, etc., are dissolved into the solution [35]. Regarding chemical food safety, Zheng et al. [29] found that phoxim pesticides on grapes can be decreased by 73% within 10 min of PAW treatment. ROS and RNS in PAW also play an important role in antimicrobial activity via rupturing of the cell membrane [32]. Therefore, PAW has been proposed as a green disinfectant for offering food safety with minimal changes in food quality [36]. Several studies investigated the inactivation efficiency of PAW on different pathogens. Suwal et al. [37] indicated that the antimicrobial activity of PAW slowly dropped during 20 min of its preparation. Several researchers reported promising results in the antimicrobial activity of PAW, including bacteria [38,39], fungus [40,41], and viruses [42,43]. Asimakopoulou et al. [44] also showed that PAW had the efficiency to inhibit both planktonic cells and biofilms of *Pseudomonas aeruginosa*, depending on the discharge frequency and exposure time.

However, there is still insufficient adoption and information on PAW generated by pinhole plasma jet to improve both food safety and quality in chili. Therefore, this research aimed to evaluate the efficacy of pesticide residue degradation (i.e., carbendazim and chlorpyrifos) and decontamination of *E. coli* and *C. gloeosporioides* using PAW. The mechanism of PAW treatments on pesticide residues, spoilage, and pathogenic microorganisms in both solution and chili were investigated. Moreover, a comparison of the efficacy of degradation and decontamination between PAW and other treatments (i.e., deionized water (DI), ethanol, and NaClO) was also performed.

## 2. Materials and Methods

### 2.1. Materials, Chemicals, and Microorganisms

Fresh chili was purchased from a local market in Chiang Mai, stored at 4 °C, and then used within 1 day. Chili was selected for uniform size (average weight and length were 5.40 ± 0.95 g and 6.50 ± 0.84 cm, respectively) and appearance without disease symptoms. Analytical standard pesticides, carbendazim (98.6% purity) and chlorpyrifos (99% purity), were purchased from Sigma-Aldrich (St. Louis, MO, USA). Nutrient broth (NB), nutrient agar, and potato dextrose agar (PDA) were purchased from Titan Biotech Ltd. (Delhi, India). All other chemicals and reagents were analytical grade from Merck (Darmstadt, Germany) and ACI Labscan (Bangkok, Thailand). Ar and O_2_ gases were obtained by Lanna Industrial Gases Co., Ltd. (Chiang Mai, Thailand). Other chemicals were of analytical grade.

*E. coli* (TISTR 527) was obtained from the Thailand Institute of Scientific and Technological Research (TISTR, Bangkok, Thailand). *C. gloeosporioides* was cultured from chili which was placed in a moist chamber at 25–28 °C until the sign of disease appeared. After spore mass appearance, chili was washed with sterilized water to collect spore mass and suspension was examined using a compound microscope (CX23, Olympus, Tokyo, Japan). Spores were then further cultured on PDA at 37 °C for 24 h. The fungus was isolated from decayed Nam Dok Mai mangoes and purified by hyphal tip isolation [26].

### 2.2. Plasma Activated Water (PAW) Preparation

The PAW was prepared using a designed pinhole plasma jet system at the Agriculture and Bio Plasma Technology Laboratory, Science and Technology Park, Chiang Mai University. The system schematic is presented in Figure 1. The system had four major parts, including (1) a pinhole plasma jet discharge system driven by a neon transformer (15 kV, 125 W) to generate plasma; (2) a gas-transport system to provide controlled flows of Ar and O_2_ gases; (3) control devices; and (4) a reservoir tank containing DI water. Each plasma jet unit has an array configuration (inset-top left), including inner coaxial tungsten electrodes and a quartz tube with an inner diameter of 0.7 mm and an outer diameter of 1.0 mm. An aluminum blade was used as a cathode and a tungsten electrode was used as anode in this system. Various radical species generated by jet plasma were collected as activated water. Accordingly, circulating activated water was carried out to increase the concentration of hydroxyl radicals (˙OH) and enhance diffusion efficiency into water. The most powerful species are the ˙OH. The mechanism of ˙OH generation by the jet plasma is summarized by Equations (1)–(4) as mentioned by Matsui et al. [44]. Therefore, a proper ratio of mixed oxygen such as 1% leads to an increase in the concentration of ˙OH species through diffusion.
e− (or Ar_m_) + H_2_O → e− + ˙OH + H˙(1)
e− (or Ar_m_) + O_2_ → e− + O(^1^D) + O(^3^P)(2)
O(^1^D) + H_2_O → 2OH˙(3)
O(^3^P) + H_2_O → 2OH˙(4)

The H_2_O_2_ concentration of PAW produced by different mixture rates of Ar + O_2_ was measured as presented in Appendix A. Thus, the best parameter setting for the highest H_2_O_2_ generation was used. For each treatment, the gas mixtures of Ar and 2% O_2_ were passed through pinhole plasma jet discharge in order to produce plasma activated water. The flow rate of Ar gas was set at 10 L/min for 6 h. The temperature for all treatments was 35–38 °C. According to this setting parameter, H_2_O_2_, which is one of the key radicals for pesticide decontamination and pathogen elimination, was generated at approximately 369.12 mg/L.

### 2.3. Efficiency of PAW for Pesticide Elimination

#### 2.3.1. In Solution

Carbendazim solution was prepared by adding 0.01 g of the standard powder into 10 mL of acetonitrile. Chlorpyrifos solution was prepared by adding 0.01 g of standard powder into 10 mL of ethyl acetate. Then, the pesticide solutions were added separately in 4 medium samples (50 mL of PAW, 500 mg/L sodium bicarbonate (NaHCO_3_), 12.5 mg/L NaClO, and sterilized DI water (control), respectively) to a final concentration of 10 mg/L. The treatment times were 5, 15, 30, and 60 min. Then, each sample was further measured in pesticide residues.

#### 2.3.2. In Chili

The pesticides (10 mg/L of carbendazim or chlorpyrifos) were prepared following the method of Phan et al. [26] with slight modification. One percent of Tween 20 as a surfactant was added to increase the adhesion of pesticides on the chili surface. Then, each pesticide solution was sprayed 3 times onto both sides of the chili sample under a fume hood (WZ1200FA, WiZard, Bangkok, Thailand) and dried with an electric fan (AVF-006, AIKO, Chon Buri, Thailand).

Then, preliminary trials for the ratio of chili and mediums (1:2, 1:4, and 3:4) and treatment time (15, 30, 45, and 60 min) were carried out. The most effective condition to eliminate both carbendazim and chlorpyrifos was the ratio at 1:2 of chili and PAW and treated for 30 min (Appendix A). Therefore, this condition of PAW was chosen for the chili treated with DI water, NaHCO_3_, and NaClO. Untreated chili was used as the control sample.

### 2.4. Efficiency of PAW for E. coli Decontamination

#### 2.4.1. In Suspension

The stock culture was initially streaked onto nutrient agar and incubated at 37 °C for 24 h. Then, a single colony was inoculated in NB and further incubated at 37 °C for 12 h. After that, the concentration of bacterial suspension was adjusted to 7 Log CFU/mL in 0.85% NaCl which was used as the working inoculum. *E. coli* was immersed in 3 medium samples (PAW, 500 mg/L of NaHCO_3_, and sterilized DI water) and treated for 15, 30, and 60 min. One milliliter of each sample was then collected after the corresponding treatment time. Due to cost and time effectiveness, the drop plate method was selected and applied to determine *E. coli* populations [45]. The detection limit was 1.50 Log CFU/mL following the method of Ekonomou et al. [46].

#### 2.4.2. In Chili

The chili sample was cleaned with 70% of ethanol and followed with 5% NaClO to reduce the microbial load before surface inoculation. The chili was further washed with DI water to remove any remaining ethanol and NaClO residues and allowed to dry in a laminar flow cabinet (LCB-903B-B2, LabTech, Jakarta, Indonesia) for 1 h before inoculation [47]. The cleaned chili was immersed in *E. coli* suspension at a concentration of 8 Log CFU/mL for 30 min. After that, the initial population of *E. coli* on the chili surface was measured and it was 6.7 Log CFU/g. To investigate the efficiency of treatment solutions (PAW, NaHCO_3_, and DI water) for *E. coli* reduction, the different ratios of chili: treatment solution were also tested. Chili samples of 25, 50, and 75 g were transferred to 100 mL (corresponded to 1:4, 1:2, and 3:4) of 3 medium samples (PAW, 500 mg/L of NaHCO_3_, and DI water, respectively) and then treated for 30, 45, and 60 min. After that, 10 g of each treated chili was added to 90 mL of 0.85% sterilized NaCl and shaken vigorously for 1 min. Then, each treatment solution was analyzed for *E. coli* by the drop plate method [45].

### 2.5. Efficiency of PAW for Decontamination on C. gloeosporioides

#### 2.5.1. Mycelial Growth

Fungal culture discs were prepared following the method of Phan et al. [26] and transferred into 4 medium samples (PAW, 10% NaClO, 70% ethanol, and sterilized DI water) and then treated for 5, 15, 30, and 60 min. Then, the fungal discs were transferred onto PDA and incubated at 30 °C for 7 days. The colony growth and hyphae characteristics of fungi after each treatment were observed using the compound microscope. The mycelial growth inhibition (%) was calculated according to Equation (5).
(5)$$\%\;\mathrm{inhibition}=\frac{\left(\left(R_{1}-R_{2}\right)\right)}{R_{1}}\times 100$$
where *R*_1_ is the diameter of the spore colony in the control group, and *R*_2_ is the diameter of the spore colony in the treated group.

#### 2.5.2. Spore Germination

One hundred microliters of fungal spore suspension (spore concentration: 10^4^ spore/mL) were transferred to an Eppendorf tube containing 500 µL of 4 different medium samples (PAW, 10% NaClO, 70% ethanol, and sterilized DI water). The spore germination was observed using the compound microscope after 48 h incubation at 30 °C.

#### 2.5.3. Anthracnose in Chili

The cleaned chili was immersed in fungal spore suspension at the concentration of 10^4^ spore/mL for 15 min and air dried by placing it on sterilized blotting paper at 30 °C for 1 h. After that, the immersed samples were transferred to the corresponding medium samples (PAW, 10% NaClO, 70% ethanol and sterilized DI water), and then treated for 5, 15, 30, and 60 min. Treated chili samples were placed in a plastic box and incubated at 30 °C for 7 days to observe the disease incidence.

### 2.6. Hydrogen Peroxide Measurement

The concentration of H_2_O_2_ was measured immediately after PAW generation as described by Phan et al. [48]. The chemicals, including 3 mL of PAW, 1 mL of 20 mg/L of KI solution, and 1 mL of 0.002 mol/L HCl, were mixed and shaken vigorously until a yellow color was observed. Then, 0.5 mL of 0.1 mg/L toluidine blue indicator was added, followed by the addition of 2 mL of 0.002 mol/L sodium acetate solution and mixed well with a vortex mixer (G560E, Scientific Industries Inc., Bohemia, NY, USA). The reaction solution was measured using a UV-Vis spectrophotometer (Genesys 10S, Thermo Fisher Scientific, Waltham, MA, USA) at 612 nm against a reagent blank. The concentration of H_2_O_2_ was calculated using linear regression from a standard curve obtained from plotting the absorbance of H_2_O_2_ solutions in the range of 0–60 mg/L (Appendix A).

### 2.7. Pesticide Residues Measurement

Chili samples were extracted using a QuEChERS Extract Pouch (Agilent Technologies, Santa Clara, CA, USA). The pesticide residues were measured following the method of Phan et al. [26] with slight modification. Carbendazim residue was analyzed using an HPLC (model 1100 Series, Agilent Technologies Canada Inc., Mississauga, ON, Canada) equipped with a C18-column (4.6 mm × 150 mm, 5 μm; Restek, Lisses, France). For chlorpyrifos residue, the residue was analyzed using gas chromatography with a flame Photometric detector (GC-FPD) (GC-6890; Agilent Technologies, Santa Clara, CA, USA) equipped with an HP-5 capillary column (30 m × 0.32 mm, 0.25 μm; Agilent Technologies, Santa Clara, CA, USA) with a split ratio of pulsed spitless. Helium was used as the carrier gas with a flow rate of 1.0 mL/min. Standard curves for carbendazim and chlorpyrifos were established using the pesticide standard solutions ranging from 0.625 to 5 mg/L. The coefficients of determination (*R*^2^) were 0.988 and 0.999 for carbendazim and chlorpyrifos, respectively. The pesticide residue concentrations were expressed as mg/kg, and the degradation efficiency of pesticides was determined as the reduction (%).

### 2.8. Statistical Analysis

The experiment was designed using a completely randomized design (CRD). All described experiments were performed in triplicate. The normal distribution of data was proven using Levene’s test. The data were analyzed for statistical significance using one-way analysis of variance (ANOVA) followed by Tukey’s multiple comparison tests (*p* < 0.05). The statistical software package (Version 16.0, SPSS Inc., Chicago, IL, USA) was used in the analysis of experimental data.

## 3. Results and Discussion

### 3.1. The Efficiency of PAW for the Degradation of Pesticides

#### 3.1.1. PAW Treatment for the Pesticides in Solution

As seen in Figure 2, degradation of about 57% of carbendazim was obtained for PAW treatment after 30 min, while treating the sample with PAW for 60 min, achieved a reduction of around 54%. PAW treatment showed significantly greater (*p* < 0.05) elimination of carbendazim and chlorpyrifos than other treatments (DI water, NaClO, and NaHCO_3_) at the same treatment time. The carbendazim treated with PAW decreased rapidly from 100% to 43.67% after 30 min of treatment, while in the samples treated with NaHCO_3_ solution, NaClO and DI water, carbendazim remained at 63.0%, 62.7% and 93.0% for the same treatment time, respectively (Figure 2A). The degradation pathway (Figure 3) of carbendazim by ˙OH has been proposed by Bojanowska-Czajka et al. [49]. Some chemical bonds of pesticides were broken by reactions from free radicals, resulting in less harmful or harmless compounds [50]. In the case of the carbendazim degradation mechanism, the ˙OH is the active radical that attack the aromatic ring and the hydrogen atoms of the methoxy group [51]. To compare with other chemical treatments, Calvo et al. [52] stated that potassium permanganate and sodium hypochlorite were ineffective for pesticide degradation due to dissolution. Ozone is another alternative to pesticide residue degradation. Ozone in the form of aqueous (ozone generally reacts with OH^−^ and H_2_O to generate H_2_O_2_) led to the degradation of pesticide residues through hydrolysis, photolysis and reduction-oxidation [51], while the radicals in PAW attack the aromatic ring of carbendazim parent structure [49].

The chlorpyrifos treated by PAW decreased rapidly from 100% to 46.1% after 60 min treatment time, while for the sample treated with NaHCO_3_ solution, NaClO, and DI water, the pesticide residue remained at about 71.8%, 53.16%, and 99.4%, respectively (Figure 2B). The mechanism of chlorpyrifos degradation (Figure 4) was proposed by Yehia et al. [53]. The P=S bond and C-O bond were broken by the reaction of ˙OH.

#### 3.1.2. PAW Treatment for the Pesticides in Chili

Figure 5 shows the reduction of carbendazim and chlorpyrifos in chili treated with PAW compared to untreated chili, DI water, NaHCO_3_, and NaClO. PAW treatment removed 80.49% carbendazim from chili. After PAW treatment, the final concentration of carbendazim (1.11 mg/kg) was below MRL (2 mg/kg) for chili according to the Codex Alimentarius [54] (Figure 5A). Treatments with DI water, NaHCO_3_, and NaClO could only reduce by 69.07%, 57.29%, and 66.43%, respectively, when compared to untreated chili. Similar results were illustrated by Ali et al. [55]. They reported that PAW using Ar/O_2_ treated on tomato fruit has the efficiency to degrade fungicides chlorothalonil and thiram residues to 75.07% and 65.89%, respectively.

For the degradation of chlorpyrifos, it was found that PAW treatment was the most efficient method to eliminate this pesticide on chili. In comparison with other treatments, the reduction of treated chili samples with PAW (65.05%) was better than NaClO (39.39%), NaHCO_3_ (25.05%), and DI water (5.86%) at the same ratio of 1:2 after 45 min of treatment time. The final concentration of chlorpyrifos (1.73 mg/kg), presented in Figure 5B, was below MRL of 2 mg/kg [56]. The degradation efficacy of carbendazim and chlorpyrifos was different due to the chemical structures of pesticides [25,57]. Higher degradation of pesticides in chili was observed. The difference between cohesive force in solution and adhesive force of pesticide on chili surface may result in better degradation. Gavahian et al. [58] also indicated that the thickness and roughness of fruit and vegetable surfaces led to greater pesticide degradation.

### 3.2. The Efficiency of PAW for Decontamination of E. coli

#### 3.2.1. Decontamination of *E. coli* in Suspension

The influence of PAW treatment on the viability of *E. coli* as compared to DI water and NaHCO_3_ was investigated. As seen in Figure 6, the growth of *E. coli* colonies after PAW treatment cultured on NB agar plate was less than those of DI water and NaHCO_3_. Quantitatively, total populations of *E. coli* on the PAW-treated suspensions were significantly lower (*p* < 0.05) than those on DI water and NaHCO_3_. After 15, 30, and 60 min, the populations of *E. coli* treated with PAW were decreased by about 0.41, 0.64, and 1.18 Log CFU/mL, respectively (the initial concentration of 6.31 Log CFU/mL). About 1.83 × 10^6^ and 2.25 × 10^6^ CFU/mL were still found in the solution treated with DI water and NaHCO_3_, respectively (Figure 6). Mošovská et al. [59] indicated that *E. coli* was reduced from 6.8 to 3.6 Log CFU/g after PAW treatment for 25 min. Additionally, the antibacterial effect persisted for 44 h. Our results are also compatible with various historical studies where the increase in treatment time in PAW enhanced the inactivation efficacy [32,60,61]. In the case of *E. coli* contamination, it can be proven that typical treatments used in the household, e.g., normal water or NaHCO_3_, cannot eliminate or reduce *E. coli*., but PAW could be a more effective alternative.

The treatment in PAW for 60 min seemed to be the most effective condition to eliminate *E. coli*. SEM technique was performed to prove the morphological changes of *E. coli* cells. The SEM images of *E. coli* cells treated by PAW and untreated *E. coli* cells are illustrated in Figure 7. The morphological changes in *E. coli* caused by PAW treatment were clearly observed by SEM. The untreated *E. coli* cells presented a rod-shaped, smooth surface morphology with intact and well-preserved cell walls and membranes, as shown in Figure 7A. After treatment with PAW for 60 min, *E. coli* cells presented distinguishable structural changes on the cell surface such as the appearance of rumple and holes in the surface (Figure 7B). The mechanism relating to these changes is the ˙OH anion in PAW. ˙OH anion is capable of oxidizing fatty acids which are an important component of the cell walls in several microorganisms, resulting also in breaking peptide bonds and oxidizing amino acid chains of microbial cells [62]. Kashmiri and Mankar [63] also reported that free radicals can attack directly polyunsaturated fatty acids in cell membranes and are capable of initiating lipid peroxidation, leading to their damage. Lipid peroxidation is a decrease in the membrane fluidity that alters membrane properties and can disrupt membrane-bound proteins. In addition, ROS, particularly ˙OH and hydrogen peroxide, could break the intra-molecular bonds of peptidoglycan, resulting in cell wall breakdown. The ˙OH subtracts the H atom from the alpha carbon of peptide bonds -CO-NH- from the peptidoglycan backbone of the cell wall linked with the amino acid [64].

#### 3.2.2. Decontamination of *E. coli* in Chili

The inactivation values of *E. coli* in chili sample (at a ratio of chili (g) and treatment solution (mL) of 1:4, 1:2, and 3:4) treated with PAW, DI water, and NaHCO_3_ and treated for 30, 45, and 60 min, respectively were investigated. The decontamination values of *E. coli* in chili samples by PAW, DI water, and NaHCO_3_ as a function of the treatment time are presented in Figure 8. The results indicated that PAW-treated chili (chili:PAW, 1:2) and treated for 60 min resulted in significantly (*p* < 0.05) lower *E. coli* populations than DI water-treated, and NaHCO_3_-treated samples. At the 1:4 ratio and treatment time of 60 min, *E. coli* populations on the PAW-treated chili samples decreased by 2.8 Log CFU/g as compared to the untreated chili (6.68 Log CFU/g). The PAW was found to have higher efficiency in the inactivation of *E. coli* in chili samples than DI water at the same treatment time and showed a tendency of greater microbial inactivation with increasing treatment time. Laurita et al. [65] stated that the effectiveness of PAW treatment is dependent on the size, shape, surface, and initial microbial contamination level of different fruits and vegetables, as well as the mode and parameter setting of PAW generation. Besides microbial inactivation on the surface of fruits and vegetables, liquid plasma was applied to the surface of pork and chicken for reducing the bacterial loads [66]. In addition, Alenyorege et al. [67] studied the effect of different frequencies of ultrasounds on *E. coli* inactivation in Chinese cabbage. Their results showed that a fixed frequency of 40 kHz could reduce *E. coli* counts > 3 Log CFU/g without adverse effects on the studied cabbage quality parameters. Barba et al. [68] have summarized different mechanisms of mild processing technologies on microbial inactivation. Protein denaturation and phospholipid crystallization, cavitation, electroporation, and inactivated cell replication are microbial inactivation mechanisms of HPP, US, PEF, and UV-light, respectively. On the other hand, reactive species generated by PAW cause damage to the microbial membrane [61].

### 3.3. The Efficiency of PAW for Inhibition of C. gloeosporioides

The efficiency of PAW on inhibition of mycelial growth and spore germination of *C. gloeosporioides* was compared with 10% NaClO, 70% ethanol, and DI water. The mycelial growth and spore germination inhibition of *C. gloeosporioides* were obtained in Figure 9. Based on the comparison, PAW treatment had a significantly (*p* < 0.05) higher mycelial growth inhibition than the treatment with 10% NaClO and control for the same treatment time. At a treatment time of 60 min, PAW treatment could inhibit the mycelial growth of *C. gloeosporioides* by 41% and showed a tendency of greater inhibition of mycelial growth with the increasing treatment time. Similar results of cold plasma treatment on the mycelial growth were also found in four fungal strains, namely *Fusarium graminearum* HX01, *Fusarium graminearum* LY26, *Fusarium pseudograminearum*, and *Fusarium moniliforme*. Increased cold plasma treatment time led to dramatic inhibition of the mycelial growth of these fungi [69]. Go et al. [70] indicated that the key mechanism of antifungal activity was the cell membrane disruption by free radicals which were generated from nonthermal atmospheric plasma.

For the inhibition of spore germination, the PAW along with 10% NaClO and 70% ethanol showed 100% inhibition in spore germination while the DI water treatment showed no inhibition of spore germination as illustrated in Figure 10. The germ tube can be clearly seen in the sample treated with DI water. For PAW, the mechanism of the inhibition of spore germination is probably due to ROS generated by plasma. ROS in PAW could oxidize the cellular compounds leading to cell dysfunction or cell damage [69]. The action of ROS on the microbial spore, for instance, bacterial spore, results in slight shrinkage, cytoplasmic leakage, and finally breakdown of the spore membrane. The transport of ROS from the PAW into the microbial cells causes internal damage through the breakdown of deoxyribonucleic acid (DNA), the destruction of proteins and other internal components of the microbial cell [65]. In addition, Thirumdas et al. [71] reported that ROS causes lipid oxidation of the cell membrane, initiating the lipid peroxidation reaction by subtracting H atoms from the unsaturated carbon of fatty acid, resulting in the formation of malondialdehyde. The malondialdehyde is used as the marker for lipid oxidation which causes damage to DNA and results in cell apoptosis.

The disease control of Anthracnose after PAW treatments is displayed in Figure 11. The results pointed out that chili treated with PAW presented high efficiency for controlling Anthracnose in chili, reducing it by 83.33% when 60 min of treatment time was applied. PAW treatment showed a significantly (*p* < 0.05) greater disease control of Anthracnose than the treatment with 10% NaClO for the same treatment time. For chili samples treated with 70% ethanol, the result demonstrated the highest control of Anthracnose in chili at every time interval. However, the chili sample treated with 70% ethanol appeared to lose weight, be dry, and have shriveled clumps due to the dehydration reaction of ethanol, which is obviously illustrated in Figure 12. Phan et al. [47] stated that ROS generated from cold plasma led to mitochondrial damage in the fungal spore membrane and inhibited spore growth of *C. gloeosporioides*, which is in accordance with our results. Furthermore, ozone, as one of the highest oxidation potentials, is also applied to control Anthracnose. The occurrence of Anthracnose on papaya was delayed and decreased by about 40% of the disease incidence [72]. Ochoa-Velasco et al. [73] also applied the coating film using the mixtures of carvacrol and thymol to decrease the incidence of Anthracnose. However, the firmness, maturity index, and color of mango and papaya were changed. Thus, it is suggested that nonthermal processing provides promising results on pesticide degradation, microbial inactivation, disease resistance, and shelf-life extension with a smaller effect on the food qualities [23,74,75].

## 4. Conclusions

PAW generated using gas mixtures of Ar and 2% O_2_ at a flow rate of 10 L/min produced the highest concentration of H_2_O_2_. The degradation of carbendazim and chlorpyrifos in both solution and chili was investigated using the optimal condition of producing the highest hydrogen peroxide concentration. PAW could effectively remove pesticides in both solution and chili. Both pesticide levels were below MRL which showed PAW effectiveness in decontaminating pesticides on chili. Besides the discovery of pesticide degradation efficiency, PAW also has the potential for decontaminating *E. coli*. The morphology of mycelial growth clearly illustrated that *C. gloeosporioides* was inhibited and controlled by PAW treatment. It was confirmed that PAW was effective in decontaminating both pesticides and microorganisms. This technology can be adopted in the production line. For instance, PAW can be used as a sanitizer to prepare raw materials (i.e., fruits and vegetables). Nonetheless, the plasma system used in the research still requires an improvement in its power and rate of PAW generation to support the application for industrial and consumer use. In addition, the effects of PAW treatment on the shelf-life of chili as well as the sensory quality of the products, and the mechanism of PAW on different products’ surfaces must be further investigated.

## Figures and Tables

**Figure 1 foods-11-02859-f001:**
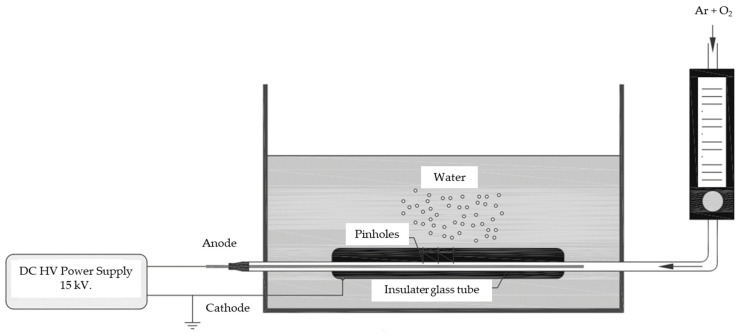
Schematic diagram of pinhole plasma jet PAW operating at atmospheric pressure.

**Figure 2 foods-11-02859-f002:**
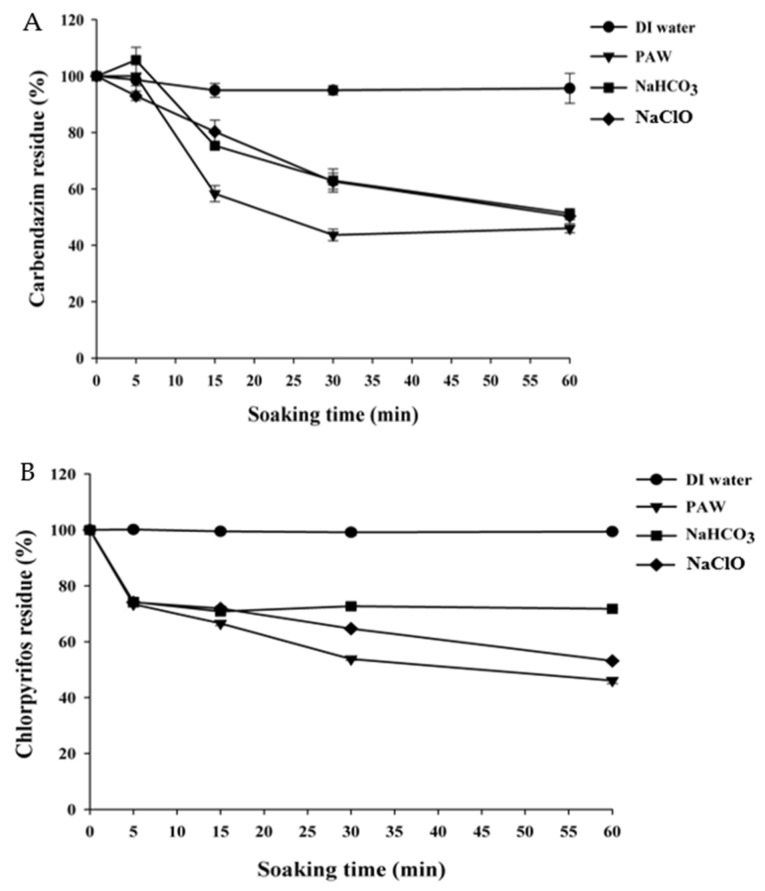
Degradation of (**A**) carbendazim and (**B**) chlorpyrifos residues in solution treated by DI water, PAW, NaHCO_3_, and NaClO as a function of treatment time.

**Figure 3 foods-11-02859-f003:**
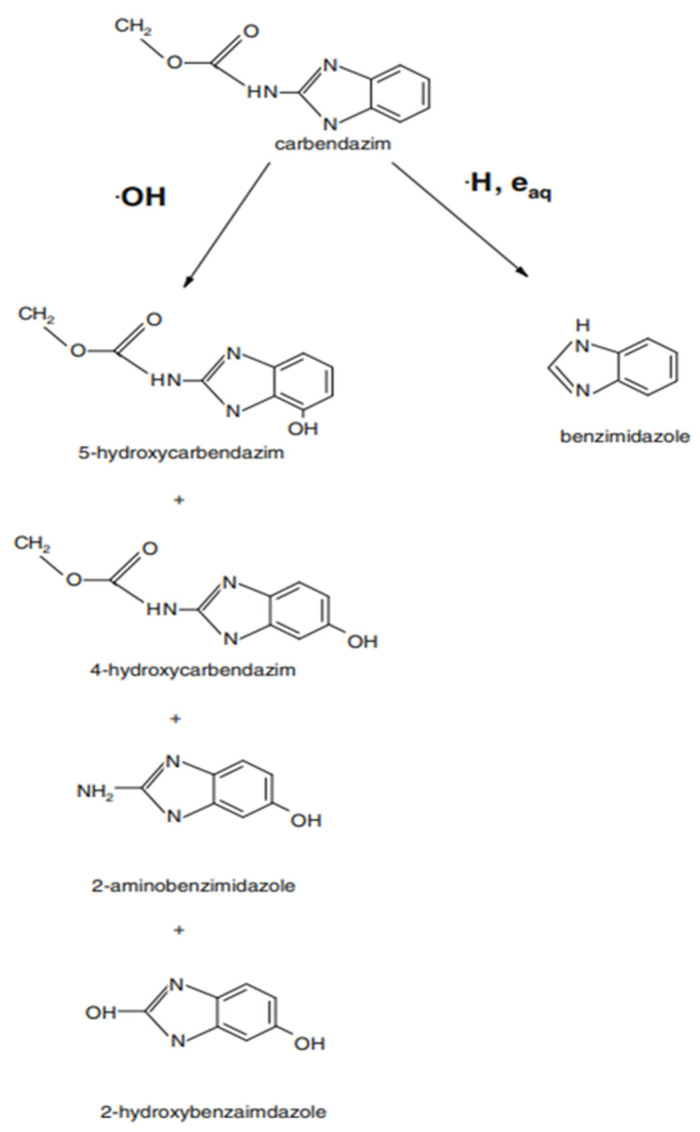
Carbendazim degradation pathway proposed by Bojanowska-Czajka et al. [49].

**Figure 4 foods-11-02859-f004:**
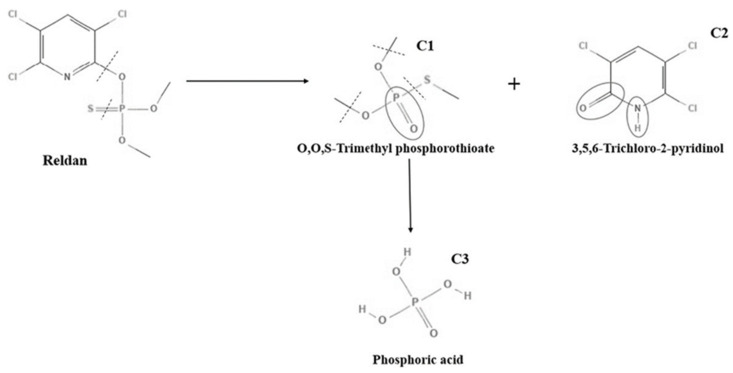
Chlorpyrifos degradation pathway proposed by Yehia et al. [53]. Obtained permission, license number: 5386981096650.

**Figure 5 foods-11-02859-f005:**
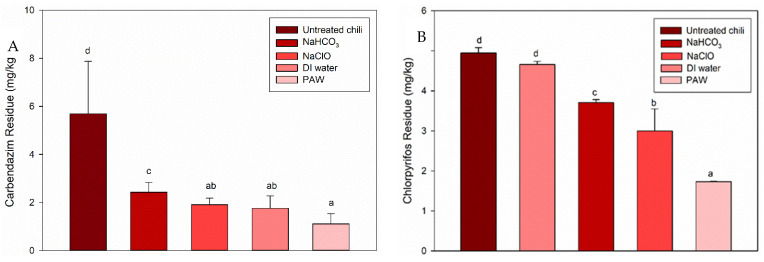
The residues of (**A**) carbendazim, and (**B**) chlorpyrifos in chili for untreated chili and treated with NaHCO_3_, NaClO, DI water, and PAW. Error bars indicate standard deviation (*n* = 3). Different letters indicate the significantly different among treatments (*p* < 0.05).

**Figure 6 foods-11-02859-f006:**
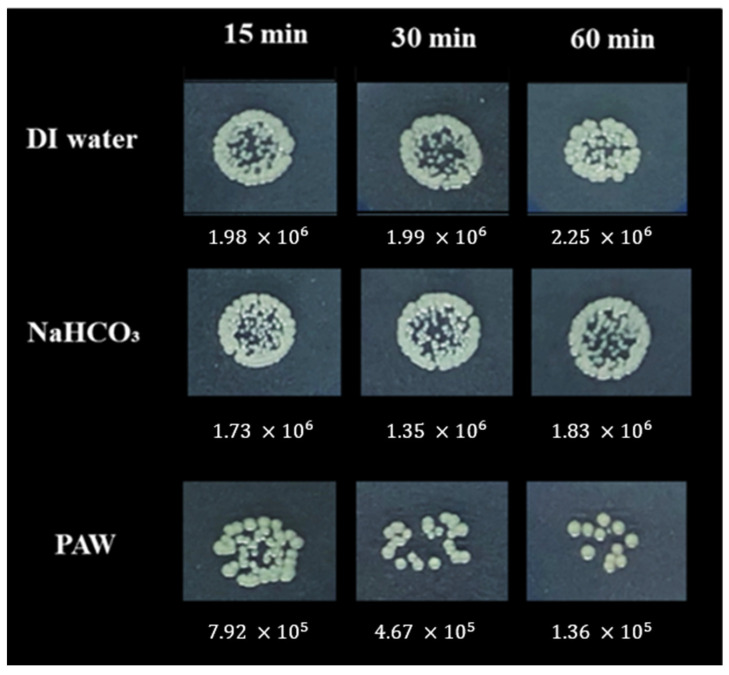
The inactivation of *E. coli* in the suspension after being treated by DI water, NaHCO_3_, and PAW after treatment time for 15, 30, and 60 min.

**Figure 7 foods-11-02859-f007:**
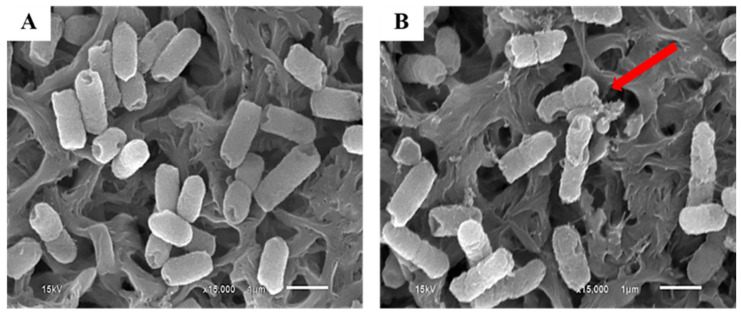
SEM images of *E. coli* cells for (**A**) untreated PAW, and (**B**) treated by PAW. Arrow indicates the rapture of *E. coli* cells after PAW treatment.

**Figure 8 foods-11-02859-f008:**
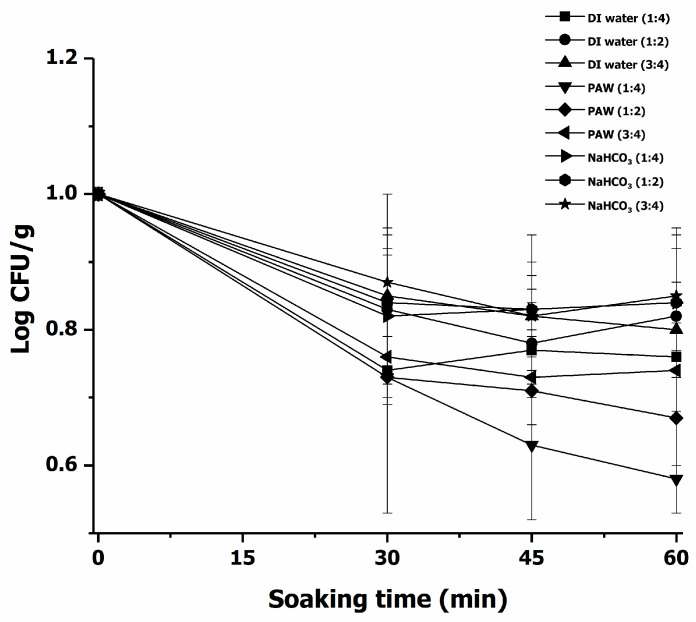
Effect of ratio of chili and mediums (DI water, PAW, and NaHCO_3_) on *E. coli* inactivation with different treatment time in chili.

**Figure 9 foods-11-02859-f009:**
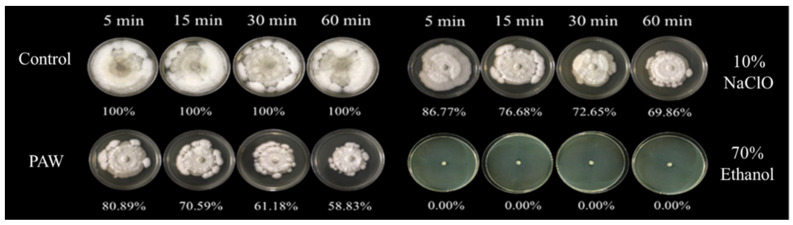
Mycelial growth of *C. gloeosporioides* after being treated with DI water, PAW, 10% NaClO, and 70% ethanol.

**Figure 10 foods-11-02859-f010:**
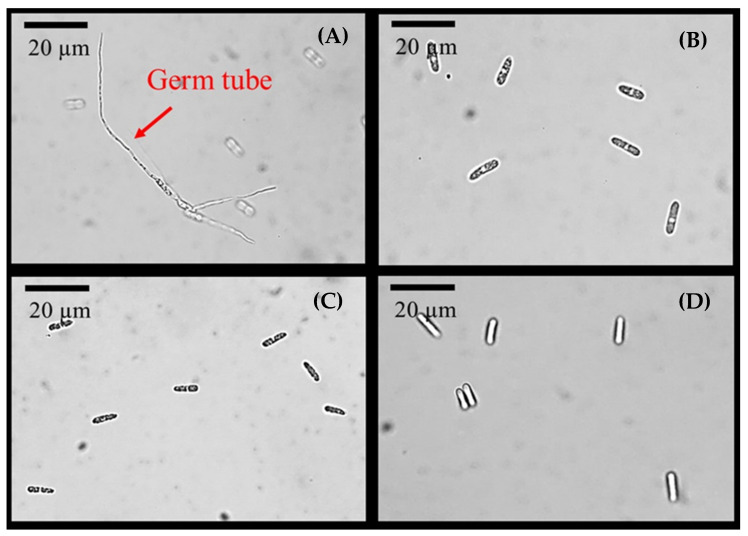
*C. gloeosporioides* spore germination after being treated for 48 h by (**A**) DI water, (**B**) PAW, (**C**) 10% NaClO, and (**D**) 70% ethanol.

**Figure 11 foods-11-02859-f011:**
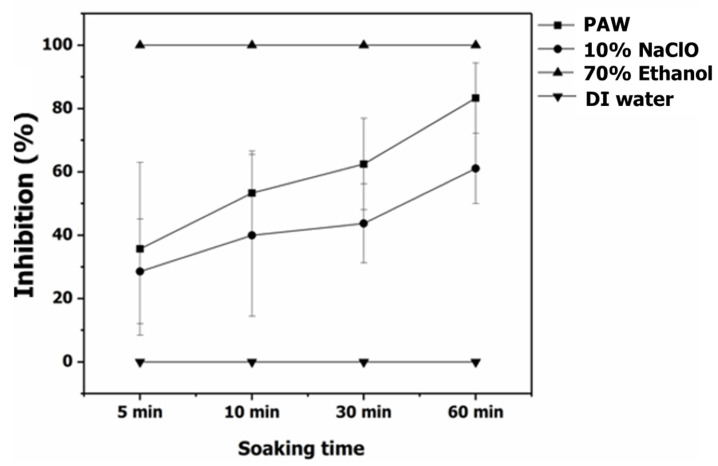
Disease control of Anthracnose after PAW treatments, 10% NaClO, 70% ethanol, and DI water.

**Figure 12 foods-11-02859-f012:**
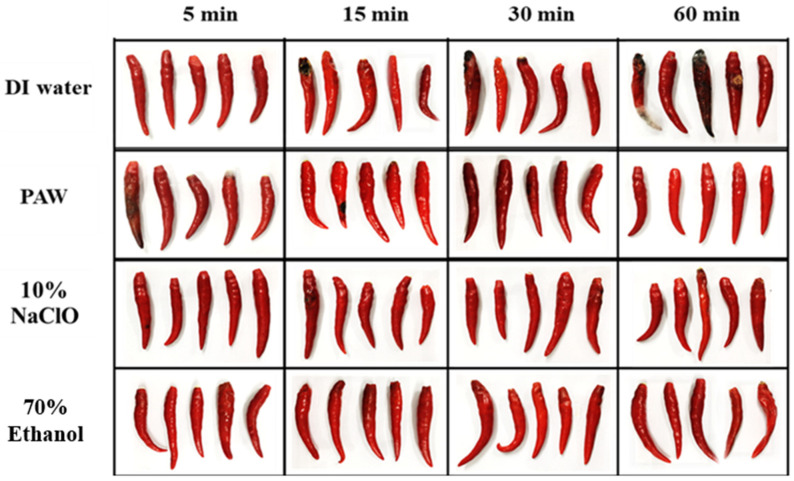
Disease control of Anthracnose in chili sample after treated by DI water, PAW, 10% NaClO, and 70% ethanol at 5, 15, 30, and 60 of treatment time.

## Data Availability

Data is contained within the article or Appendix A.

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
