# Peer review of "Application of Pinhole Plasma Jet Activated Water against Escherichia coli, Colletotrichum gloeosporioides, and Decontamination of Pesticide Residues on Chili (Capsicum annuum L.)"

_foods, 2022, doi:10.3390/foods11182859_

Round 1

Reviewer 1 Report

The paper investigates the decontamination efficiency of PAW against Escherichia coli and Colletotrichum gloeosporioides that can be found on chilli peppers.

The paper is interesting, but significant changes need to be made to help the authors straightforwardly describe the main findings on the application of PAW treatment as a new sanitization method.

Reviewer 2 Report

Dear Authors,

Detailed notes on the manuscript are as follows:

1) In the introduction, it should be added that food safety also uses technologies based on UV-C, microwaves, magnetic field, etc.

1a) DOI 10.15199 / 48.2020.01.55, DOI 10.1051 / bioconf / 20181002031, DOI10.3390 / su12083426

2) Improve the readability of figure 1 (description in the figure)

3) 2.8 Statistical analysis - the use of Tukey's test indicates that the ANOVA was parametric (add information about the study of the normal distribution of the data population, homogeneity of variance in samples; what tests - it is customary to include the results of these tests in the result part)

4) Figure 2. Degradation of…. - data on the chart should not be combined, it indicates data prediction (leave a spread of points or insert a trend line + coefficient of determination)

5) Figure 5. The residues of ... - add what the error bars mean (Sd, 5%, mean, etc.), a-d homogeneous groups

6) Figure 8. Effect of… - see pt. 4

7) Figure 10.- arrange the pictures 2x2 and enlarge (they are very important, they should be legible)

8) Figure 11. Disease control…. - see point 4 (follow MDPI standard)

9) Conclusions - "... It could be seen that PAW was effective in decontaminating both pesticides ..." (line 451) - develop the application, indicate possible application applications, show that it works and can be used in industry.

Round 2

Reviewer 1 Report

The submitted manuscript has been improved significantly. However, some minor changes need to be considered before publication in Foods. Finally, I would suggest the authors go through the whole manuscript in detail and make some changes. I noticed quite a few typos and problems with the English language. 

Reviewer 2 Report

Dear Authors, thank you for making changes. I accept your explanations.
